# Fabrication and Characterisation of Organic EL Devices in the Presence of Cyclodextrin as an Interlayer

**DOI:** 10.3390/s21113666

**Published:** 2021-05-25

**Authors:** Michihiro Hara, Takao Umeda, Hiroyuki Kurata

**Affiliations:** Department of Applied Chemistry and Food Sciences, Faculty of Environmental and Information Sciences, Fukui University of Technology, Gakuen 3-6-1, Fukui 9108505, Japan; 2027@edu.fukui-ut.ac.jp (T.U.); kurata@fukui-ut.ac.jp (H.K.)

**Keywords:** organic EL, cyclodextrin polymer, interlayer, energy saving applications

## Abstract

This study examined glass-based organic electroluminescence in the presence of a cyclodextrin polymer as an interlayer. Glass-based organic electroluminescence was achieved by the deposition of five layers of N,N’-Bis(3-methylphenyl)N,N’-bis(phenyl)-benzidine, cyclodextrin polymer (CDP), tris-(8-hydroxyquinolinato) aluminium LiF and Al on an indium tin oxide-coated glass substrate. The glass-based OEL exhibited green emission owing to the fluorescence of tris-(8-hydroxyquinolinato) aluminium. The highest luminance was 19,620 cd m^−2^. Moreover, the glass-based organic electroluminescence device showed green emission at 6 V in the curved state because of the inhibited aggregation of the cyclodextrin polymer. All organic molecules are insulating, but except CDP, they are standard molecules in conventional organic electroluminescence devices. In this device, the CDP layer contained pores that could allow conventional organic molecules to enter the pores and affect the organic electroluminescence interface. In particular, self-association was suppressed, efficiency was improved, and light emission was observed without the need for a high voltage. Overall, the glass-based organic electroluminescence device using CDP is an environmentally friendly device with a range of potential energy saving applications.

## 1. Introduction

Organic electroluminescence (OEL) is the feature of a light-emitting device that utilises fluorescence or phosphorescence emitted from an organic thin film. The device can be made thinner than a liquid crystal display device because a backlight is unnecessary. Further features include surface light emission, high image quality and high-speed response, permitting applications in next-generation flat-panel displays. OELs require high-performance monochrome or white-light-emitting materials. Relevant studies have been conducted on the synthesis and physical properties (e.g., aggregation-induced emission and thermally activated delayed fluorescence) [1,2,3,4,5]. Although OELs are generally manufactured using glass substrates, investigations have also included the fabrication of film-type OEL devices, considering flexibility without being limited to flat surfaces [6,7]. In recent years, the concept of flexible OEL smartphones and OEL TV devices has been reported, and the installation of OELs is expected to become common in the future [8,9,10,11,12]. Meanwhile, OEL is expected to attract attention as a subject for energy saving and environmentally friendly devices in future lighting and display applications. Furthermore, as a necessity for energy saving, low-voltage driving is an important condition for reducing power consumption.

Cassandre et al. synthesised 4-phenyl-N-carbazole-spirobifluorene and 4-(3,4,5-trimethoxyphenyl)-spirobifluorene that exhibited green (external quantum efficiency (EQE) = 20.2%) and blue (EQE = 9.6%) phosphorescent-type OEL, respectively [13]. Sylvain et al. described white-light emission by controlling the protonation of an electron donor (D)-π- electron acceptor (A) push–pull molecule comprising a methoxyphenyl/methoxynaphthyl group as D and diazine as A [14]. Further reports also dealt with, among others, a white-light-emitting OEL using the exciplex light emission of a hole-transport material and a light-emitting material [15,16,17]. In addition to the synthesis of new materials, research on the fabrication of OELs that exhibit high-efficiency light emission has also focused on topics such as the control of quenching of excited states between the hole-transport and light-emitting layers, and the promotion of charge recombination. Examples of relevant reports include the insertion of an interlayer between the hole-transport and light-emitting layers, control of excited triplet state quenching, and enhanced efficiency by blocking electrons (e^−^) passing through the light-emitting layers [18,19,20,21].

This study examined the fabrication and characterisation of low-voltage-driven OELs. Cyclodextrin (CD) was used as an interlayer. CD is a series of oligosaccharide homologs with excellent crystallinity, having a structure in which a number of D-glucopyranoses are cyclised via α-1,4 glycosidic bonds to form a crown shape. According to the number of glucose units constituting CD, these are called α-CD (six units), β-CD (seven units), and γ-CD (eight units). CDs are hydrophobic inside the ring and hydrophilic outside. These compounds containing CD are used widely in the food and pharmaceutical industries because they are stable and do not easily oxidise or evaporate. In this respect, considering the molecular stability of CD, a polymer of CD (CDP) has been used, such as in the development of the interlayer composite photocatalyst and the operation of the aqueous solutions inclusion complex as a new polymer material, respectively [22,23]. This CDP is obtained by removing the H of the hydroxyl group of the glupyranose group and bonding multiple hydroxyl groups of another glupyranose group to form a three-dimensional crosslinked polymer. This polymer is insoluble in water and organic solvents while maintaining the properties of CD. The CDP has been evaluated as an iodine, polyphenols, and flavonoid adsorbent, adsorbent, water purifier and surfactant remover. Xiao Han et al. introduced the research progress into CDPs and their applications in many fields [24]. A previous study reported an improvement in the function of CD using a photoelectric conversion element, namely a ‘Dye-sensitised solar cell with CD layer’, which is a similar eco-device, which is an absorbent material, has a photochromic reaction field and a cavity size effect for the absorbed material, respectively [25,26,27]. In the present study, this CDP was selected as an interlayer, and an OEL device was fabricated for low-voltage driving for volume loss and minimisation of the next-generation device. The overall aim was to manufacture environmentally friendly OEL devices using plant-derived materials, such as CDP.

## 2. Materials and Methods

This study examined the fabrication and characterisation of low-voltage-driven OELs. Cyclodextrin (CD) was used as an interlayer.

N,N’-Bis(3-methylphenyl)-N,N’-bis(phenyl)-benzidine (TPD, Luminescence Technology Corp., Figure 1), tris(8-hydroxyquinolinato) aluminium (Alq_3_, Luminescence Technology Corp., Figure 1), LiF (Nacalai Tesque Co.) and aluminium (Nirako Co.) were placed in an evaporation boat attached to the lower portion of a vacuum deposition apparatus. Subsequently, an etched and cleaned ITO conductive plate glass substrate (Tokyo Sankyo Shinku Corp. 15 Ω/□) was installed on the upper portion of the vacuum evaporation apparatus and covered with a bell jar. A circulating cooling device (RTE-100 NESLAB) was used to cool the film thickness sensor. Liquid nitrogen was injected into the liquid nitrogen trap of the vacuum evaporation apparatus until the vacuum reached 2.0 × 10^−6^ Torr or less. Subsequently, an OEL element was formed by vacuum deposition in the following order: TPD, Alq_3_, LiF and Al. The thicknesses of the organic layers (TPD and Alq_3_), LiF and Al were adjusted to 50, 1 and 150 nm, respectively, using a crystal vibration-type film-forming controller (ULVAC, CRTM-6000). To calculate the film thickness precisely, the thickness was corrected using the needle-touching method with a stylus surface profiler (ULVAC Dektak).

OEL elements containing α-cyclodextrin polymer (α-CDP, FUJIFILM Wako Pure Chem. Co.), β-cyclodextrin polymer (β-CDP, FUJIFILM Wako Pure Chem. Co.), and γ-cyclodextrin polymer (γ-CDP, FUJIFILM Wako Pure Chem. Co.) as an interlayer (IL), were also produced. After vapour deposition of TPD, α-CDP, β-CDP, γ-CDP and Alq_3_ in the same manner described above, the bell jar was opened. The evaporation boats were removed, and other evaporation boats containing LiF and Al were installed. LiF and Al were then deposited sequentially to produce OEL devices containing various CDPs (Figure 2).

## 3. Results and Discussion

### 3.1. Characterisation of the OEL Device

Upon application of a voltage to the fabricated OEL, green device light emission (λ_max_ = 520 nm) was confirmed (Figure 3).

An energy diagram was produced to examine whether the emission from the OEL originated from the Alq_3_ light-emitting layer (Figure 4). The work function of ITO was reported to be approximately 5.0 eV after ultrasonic cleaning using a solvent and UV cleaning. Here because the highest occupied molecular orbitals of TPD and Alq_3_ are 5.5 and 5.8 eV, respectively, it is suggested that energetic injection of holes (h^+^) from the anode side into the Alq_3_ layer is possible. Regarding electron (e^−^) injection from the cathode, electrons are injected from Al (4.3 eV) to LiF (2.9 eV) and from LiF to the lowest unoccupied molecular orbital (LUMO, 3.1 eV) of Alq_3_ via the tunnel effect. However, it was confirmed that no e^−^ injection occurred from the Alq_3_ layer to the TPD layer because the LUMO of TPD is 2.4 eV. Consequently, this phenomenon is possible by the recombination of h^+^ and e^−^ in the Alq_3_ layer and exciton generation in terms of optical emission. Furthermore, according to the results of measurement of the fluorescence spectrum of the OEL, a spectrum similar to that of powder Alq_3_ was observed. These results confirm that the light-emitting Alq_3_ layer was the origin of the emission in the OEL.

The characteristics of the OEL were evaluated based on *L*–*V*–*I* characteristics (Figure 5). When a voltage of 11 V was applied, the luminance and current density of the OEL device were 19,620 cd m^−2^ and 100 mA cm^−2^, respectively. The inset shows an enlarged view. By contrast, the OEL without CDP showed a luminance of 1.9 cd m^−2^.

Note that upon introduction of CDPs into the interlayer and application of a voltage of 11 V, the fluorescence spectrum at ~550 nm of CDP-OEL approximately coincided with that of Alq_3_ (Figure 6). The λ_max_ for the OEL layers without CDP and the OEL layers with αCDP, βCDP and γCDP was 520, 540, 545 and 535 nm, respectively. Although this was not a large change, a 20-nm red shift in wavelength was observed. This means that inserting the CDP layer affects the formation of the interface between TPD and Alq_3_. Furthermore, the luminance and current density decreased to ~20–50 and ~1–2 times, respectively (Table 1). This suggests that the CDP layer acts as an insulating layer and the luminance is reduced and increased. The electrical resistance of CDP is different compared with that of TPD and Alq_3_; hence, the current may be different. The results suggest the formation of OEL using a CDP layer as the IL.

### 3.2. Low Voltage Drive-Type OEL

The effects of various CDPs as an ILs were verified by comparing their performance with that of OEL containing no CDP. The maximum luminance decreased when CDP was used as the IL (Figure 7). The difference of various CDPs as ILs was that α-CDP and β-CDP were observed to change with voltage compared to the device without CDP, and γ-CDP showed an inhibitory effect at all voltages. The effect of α-CDP and β-CDP as an IL showed the increase of IL at 5, 6, and 7 V, and an inhibitory effect at greater than 8 V. It appeared that there was a change in the IL because of the difference in resistance through the insertion of CDP, which is an insulating layer. In addition, it was shown that the effect of inclusion of organic molecules in the pores of CDP had a greater effect. Therefore, we concentrated on 6 V, which had the greatest change at this low voltage. When 6 V was applied (low voltage), the OEL device without CDP showed a luminance of 1.9 cd m^−2^, and those comprising α-CDP, β-CDP and γ-CDP had values of 8.4, 3.1 and 0.0073 cd m^−2^, respectively (Figure 7, Table 2). This confirmed the effect of the α-CDP layer as an IL. The luminance increases because in an OEL device containing α-CDP in the CDP layer, as the molecules are included in the holes one by one, the association between TPD and Alq_3_ is suppressed, and excitons are generated efficiently. The luminance did not increase when the applied voltage of the OEL device containing β-CDP and γ-CDP was 6 V. A plausible explanation is that the small pore size of β-CDP prevents the inclusion of adjacent organic substances (TPD or Alq_3_). This is similar to the association of the same organic substances and the suppression of electron transfer between identical molecules. Moreover, the luminance was lower because β-CDP worked as an insulating layer instead of acting as an IL. However, the inclusion of associated organic substances is also possible because the hole size of γ-CDP is large, and the increase in brightness is reduced because electrons are exchanged between molecules in the inclusion complex. Therefore, α-CDP was most useful as an IL.

## 4. Conclusions

To obtain TPD/Alq_3_-based OEL devices with a low voltage drive, OELs were fabricated using various CDPs (α-CDP, β-CDP and γ-CDP) with different pore sizes and characterised. The effects of differences in the CDP used in the OEL devices were also examined.

Regarding the TPD/Alq_3_-based OEL devices without CDP, a maximum luminance of 19,620 cd m^−2^ was observed at an applied voltage of 11 V. The maximum luminance of TPD/Alq_3_-based OEL devices containing various CDPs as ILs was lower than that of TPD/Alq_3_-based OEL devices containing no CD. However, improved luminance was observed upon the application of a low voltage (6 V). At a low driving voltage (applied voltage of 6 V), the OEL device without CD showed a luminance of 1.9 cd m^−2^, whereas the OEL devices with various CDPs as the IL, and containing α-CDP, β-CDP and γ-CDP, showed luminances of 8.44, 3.07, and 0.073 cd m^−2^, respectively. A comparison of the luminance of the TPD/Alq_3_-based OEL devices containing CDP revealed a difference in luminance due to the pore size (molecular size), with α-CDP (6Å) showing the highest luminance. These results confirm the possibility of driving a TPD/Alq3-based OEL device at a low voltage by the vapour deposition of α-CDP. Among α-CDP, β-CDP and γ-CDP as interface scaffolding, the pore size of α-CDP proved to be optimal.

It was determined that CDP could be used as a scaffold to eliminate each association and control the interface. In this paper, only the organic layer interface was considered; however, the LiF and ITO interface will also be investigated. Creating an OEL constant voltage dimmer makes it possible to develop energy saving devices. Furthermore, a safe CD that is derived from plants can be used in food and a real eco-friendly device can be created from the point of view of materials. Thus, this research can lead to development of an environmentally friendly device that promotes sustainability.

## Figures and Tables

**Figure 1 sensors-21-03666-f001:**
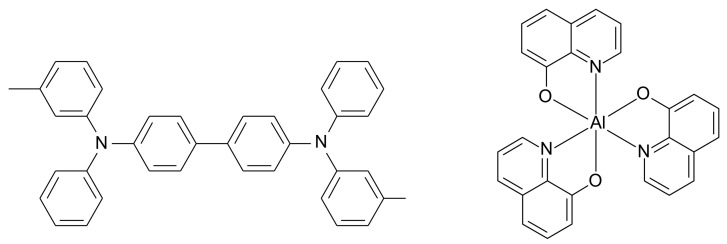
Molecular structures of TPD (left) and Alq_3_ (right).

**Figure 2 sensors-21-03666-f002:**
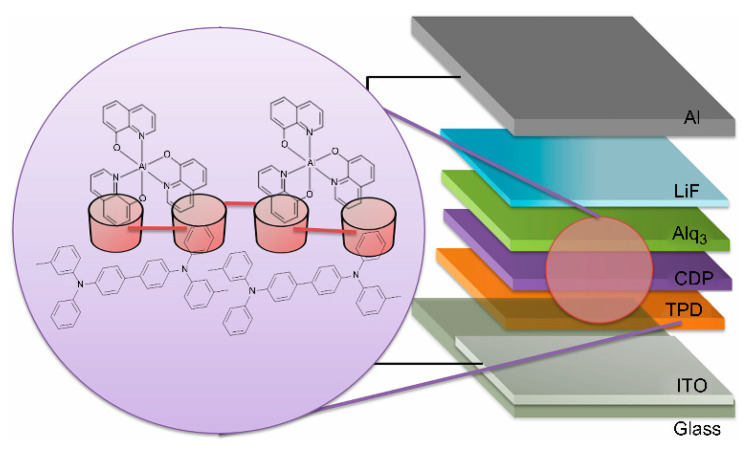
Structure of OEL in the presence of cyclodextrin as an interlayer.

**Figure 3 sensors-21-03666-f003:**
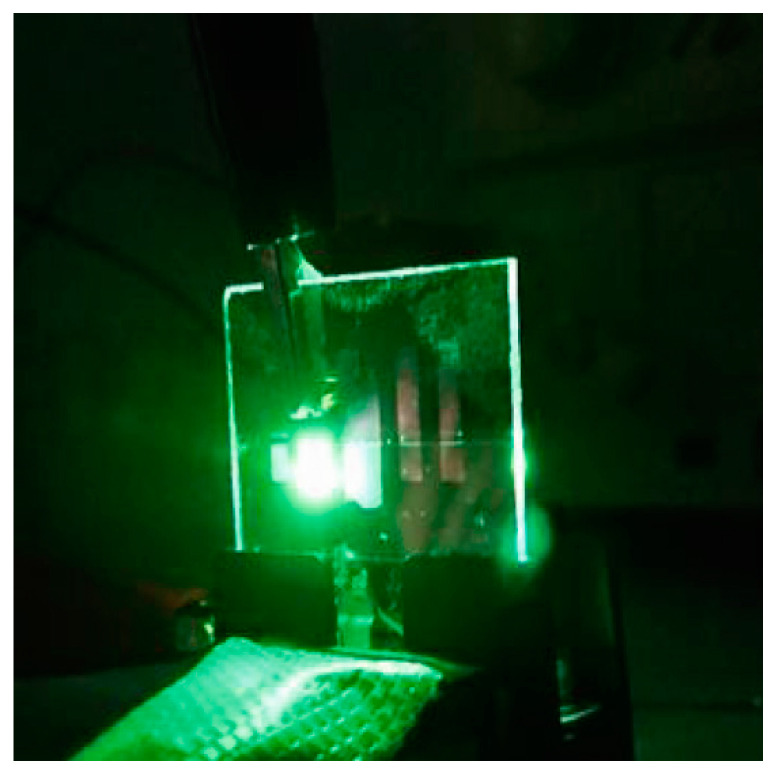
Photograph of the emission of the OEL device (TPD (50 nm)/Alp_3_ (50 nm)/LiF (1 nm)/Al (100 nm) in the curved state.

**Figure 4 sensors-21-03666-f004:**
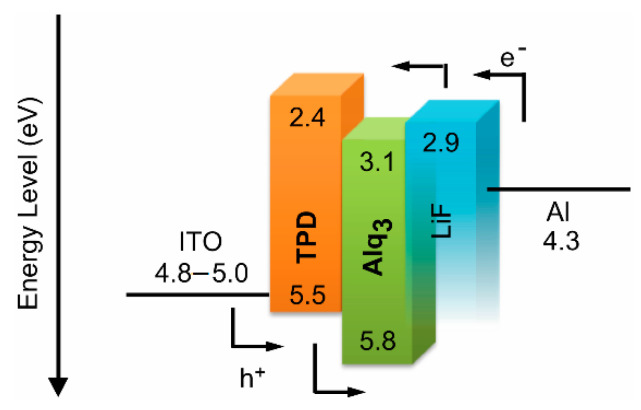
Energy level diagram of OEL.

**Figure 5 sensors-21-03666-f005:**
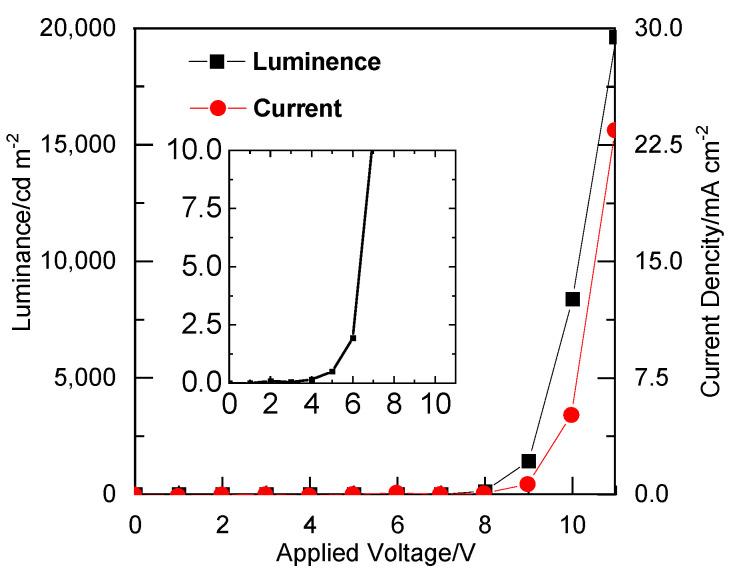
(A)*L*(■)-*V*-*I*(●) profiles of the OEL device not containing CDP. TPD (50 nm)/Alq_3_ (50 nm)/LiF (0.1 nm)/Al (150 nm). The inset shows an enlarged view.

**Figure 6 sensors-21-03666-f006:**
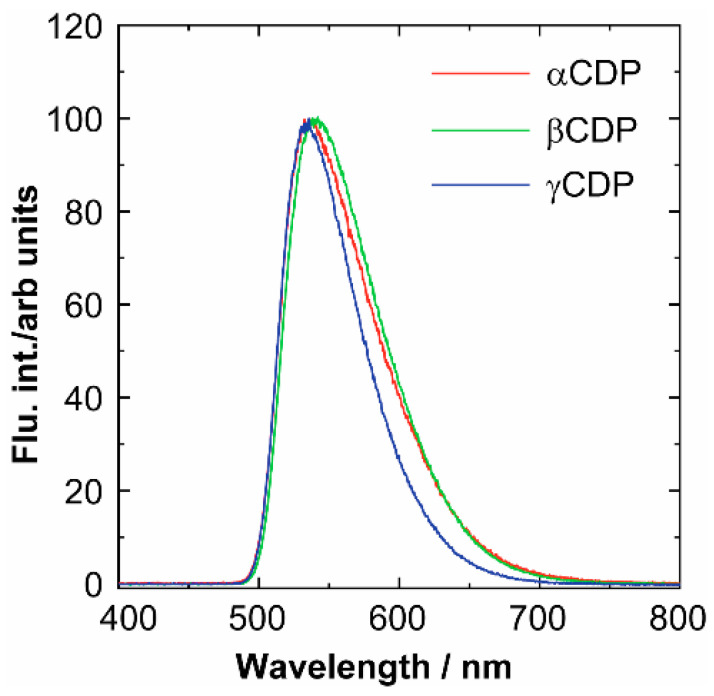
EL spectra of the OEL devices. TPD (45 nm)/CDPs (10 nm)/Alq_3_ (45 nm)/LiF (0.1 nm)/Al (150 nm).

**Figure 7 sensors-21-03666-f007:**
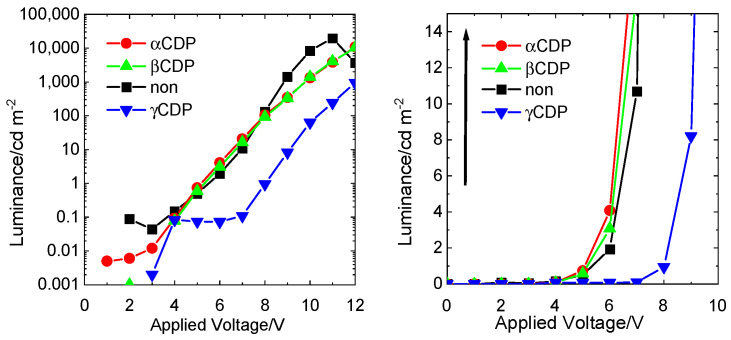
*L*-*V* profile of OEL devices containing CDPs (10 nm). Left: the applied voltage at 0–12 V vs. logarithm of the luminance intensity and right: the applied voltage at 0–10 V vs. the luminance intensity. TPD (45 nm), CDPs (10 nm), Alq_3_ (45 nm), LiF (0.1 nm), Al (150 nm). (non ■, αCDP: ●, βCDP: ▲, γCDP: ▼).

**Table 1 sensors-21-03666-t001:** Performance of OEL devices. TPD (45 nm)/CDPs (10 nm)/Alq_3_ (45 nm)/LiF (0.1 nm)/Al (150 nm).

CDP	Applied Voltage (V)	Luminance (cd m^−2^)	Current Density (mA cm^−2^)
Non	11	19,620	100
αCDP	12	10,880	103
βCDP	12	10,330	205
γCDP	14	4886	101

**Table 2 sensors-21-03666-t002:** Performance of an OEL device not donating CDPs and containing α.β.γ-CDP at an applied voltage of at 6 V. TPD (45 nm, /CDPs (10 nm), Alq_3_ (45 nm), LiF (0.1 nm), Al (150 nm).

CDP	Applied Voltage (V)	Luminance (cd m^−2^)	Current Density (mA cm^−2^)
Non	6	1.9	0.11
αCDP	6	8.4	0.02
βCDP	6	3.1	0.77
γCDP	6	7.3 × 10^−3^	7.4 × 10^−3^

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
