# Peer review of "Fabrication and Characterisation of Organic EL Devices in the Presence of Cyclodextrin as an Interlayer"

_sensors, 2021, doi:10.3390/s21113666_

Round 1

Reviewer 1 Report

This work in on the fabrication of OLEDs based on the insertion of an interlayer of

 cyclodextrin polymer. Autors claim improving the performance of the device at low voltages when the layer is present compared with those in which is not present. Based in a standard design such as TPD/Alq3/ LiF/Al, the insertion of the cyclodextrin polymer layer between TPD and Alq3 avoids, according to author’s claims, the molecular association of Alq3 and TPD. This in turn improves the device performance at low voltages such a 6 V. At higher voltages the cyclodextrin does not introduces benefits. All explanations are reasonable, and the results are of interest. However, for consideration of acceptance the following should be clearly demonstrated:

  • Based on the plot of Figure 7 is difficult to stablish the actual improvement of device with the insertion of cyclodextrin polymer as interlayer. Could authors present a L-V profile with smaller steps in the applied voltage? In the actual profile the step is 1 V such that all the claims are given with only one point that resolved all the trend. It would be interesting to see the whole trend between 5 and 6 V, and further, between 6 and 7 with more points in the middle.
  • Please justify the advantages and utility of having just weak luminance at 6 V even when cyclodextrin polymer as interlayer introduces a ca 5 fold factor
  • Introduction Ref [22-23]. Pleas clarify the motivation, application and briefly the advantages of using cyclodextrin in these references. The same for [24-26]
  • In page 7 it say that the luminance is 1.88 cd/m2 at 6 V in a device without cyclodextrin polymer as interlayer, but table 2 the value is 1.9.

Author Response

Dear Reviewer,

We thank referees for careful reading our manuscript and for giving useful comments. In response to the Referees' comments, we have revised the Manuscript ID: sensors-1198506.

We look forward to a publication of our manuscript in Sensors.

Sincerely, Michihiro HARA

Reviewer 2 Report

This manuscript by Hara et al. reports glass-based organic EL in the presence of a clodextrin polymer as an interlayer. All the methods and outperform regarding in the manufacture of environmentally friendly device is great. All in all, the quality of the manuscript is sufficient and the results are interesting. I strongly recommend this work in Sensors in this present form.

Author Response

(The authors gave the same response as above.)

Reviewer 3 Report

Thank you for your contribution. You have provided a sound comparison of the effect of CDP in your OEL devices.  

Please add the CD-polymer molecular structure and identify the variations for the reader.

In your paper you discuss the notion of pores in the CDP layer that are responsible for the EL behavior change.  Please provide evidence to support this. Could you please add some morphological or topological evaluation of the organic films in your devices or other physical evaluation?

Please review lines 15,16 in the abstract for typos.

Author Response

(The authors gave the same response as above.)

Round 2

Reviewer 1 Report

Authors have addressed my suggestions although the revised manuscript was just marginally improved. In my opinion, the luminance vs V Figure that authors attached to the response letter should be part of the manuscript, along with a broader discussion and edition of the manuscript according to my original review.

Author Response

sensors-1198506

Type of manuscript: Communication

18 MAY 2021

Dear Reviewer,

We thank referees for careful reading our manuscript and for giving useful comments. In response to the Referee re-comment, we have revised the Manuscript ID: sensors-1198506.

We look forward to a publication of our manuscript in Sensors.

Sincerely, Michihiro HARA

Our responses to the referees' reports are as follows:

Response for the Referee

1) Authors have addressed my suggestions although the revised manuscript was just marginally improved. In my opinion, the luminance vs V Figure that authors attached to the response letter should be part of the manuscript, along with a broader discussion and edition of the manuscript according to my original review.

Answer:

Thank you for your advice. As you pointed out, we inserted the diagram and reinserted the discussion.

 We have inserted Figure 7 Left and the caption, and “The difference of various CDPs as an…and then consider.”(page 6, line number 171-179).

Please address all correspondence to:

Corresponding Author: Michihiro Hara

Department of Environmental and Biological Chemistry,

Fukui University of Technology

3-6-1 Gakuen, Fukui 910-8505, Japan

E-mail: hara@fukui-ut.ac.jp

Tel.: +81-776-29-2446

Fax: +81-776-29-2446

We look forward to hearing from you at your earliest convenience.

Yours sincerely,

Prof. Michihiro Hara Dr.